# Enhancing Medical NLP Systems: Integrating Upstash Vector and BGE-M3 for Accurate and Ethical Healthcare Data Management with Reduced Bias

## Abstract

This paper proposes a novel NLP model in healthcare by including Utash Vector for in-time and contextual information retrieval and BGE-M3 for advanced understanding. The model overcomes the challenges posed by the existing systems, such as incomplete data retrieval, a semantically inconsistent database, and algorithm bias. Incorporating bias mitigation measures and fairness audits, it guarantees no unfair treatment of patients belonging to different groups. Aligned with the AMA Code of Medical Ethics, provides proper management of Electronic Health Records in better ways in terms of transparency, confidentiality, and accuracy. Although these problems are relieved, the accuracy of information is still a major issue, the abuse of artificial intelligence remains a risk, and the use of the AMA Code to guide the integration of artificial intelligence has its limitations. Each of these must operate with defensible use of AI and auditing as well as explanation of AI usage in clinical decision-making.

## 1 Introduction

Since Artificial Intelligence (AI) and Natural Language Processing (NLP) technologies are now set to be prevalent in the healthcare sector, it is anticipated in this paper that Electronic Health Records (EHR) systems will serve as useful instruments for storing information on the patients. Nonetheless, the current application of AI technology in the EHR systems does not come without limitations such as lack of data quality, bias inside the systems and ethical approaches to providing the technology. These constraints not only restrict the clinical utility of AI but also increase health inequities as well as a scepticism towards AI.

Another limitation is the phenomenon of the patients not being at the center of care, providing generic approaches and insufficient focusing for patients' needs. The current models of AI continue to incorporate a one-size-fits-all approach whereby strategies and plans advanced to the patients do not consider their unique features such as their past illnesses. Although individualization is important in making medical decisions, the current analysis models are too simplistic that even with all the intention to individualize the outputs, nothing is changed – they continue churning out the same outcome no matter the patient's condition or history (Honavar, 2020).

Moreover, insufficient appropriate procedures and complexity if archived data in EHR systems is a major drawback. Due to the target population's frequently presented reluctance to participate in future clinical trials, complete EHR data and histories are typically not available. This further delays the diagnosis and treatment of patients and raises the risk of diagnostic errors where doctors fail to manage the scarcity of time and the enormous load of data (Weiskopf & Weng, 2013).

Apart from these outcomes common in every clinical practice, the challenges of bias embedded in AI models is rather tremendous. The healthcare history reflected in EHR dataset is also fairly biased in nature. gender bias, race, etc. This means that the AI systems which are based on such data will also enforce such biases and may worsen them. Parasuraman & Manzey (2010) found that American Black patients with similar medical conditions as American White patients were provided

with fewer healthcare resources which shows managing health information is faulty and affects treatment outcomes.

Another critical limitation is the lack of ethical reasoning in the current AI systems. Ethical dilemmas are common features in decision-making processes in medical practices especially with regards to patient confidentiality, informed consent, and euthanasia. In any case, most of the AI systems are oriented to the clinical outcomes while leaving the ethical aspects necessary for full and patient-oriented care (Code of medical ethics) unattended. Such lack of embracing such ethics in AI exacerbates the problem at hand, especially when it comes to treatment matters concerning patients.

To counter these limitations, this paper argues for better reforms within this particular NLP considering the inclusion of the American Medical Association (AMA) Code of Medical Ethics and EHR. This is meant to ensure that therapeutic recommendations provided by the AI are clinically accurate and ethically defensible as well as personalized by addressing bias through the BGE-M3 model. Utilizing coercive techniques to influence behavior opinion change. Upstash Vector, a fast-growing and efficient vector database will enhance the search capacity of both clinical and ethical components so as to provide personalized, directed advice in real time. As this method will deal with various aspects such as bias, ethical reasoning, data complexity, and personalization, it will result in improved health and well-being that will enhance equity and transparency.

## 2 CURRENT CHALLENGES

### 2.1 FRAGMENTED DATA AND LACK OF INTEROPERABILITY IN EHR SYSTEMS

Electronic Health Records (EHR) systems have advanced healthcare by digitalizing patient data, yet the system faces significant challenges. One of the main issues is that the data contained in the EHRs is unsystematic and is mostly free text comprising of clinical notes, physician notes, and other documents which present a challenge to Natural Language Processing (NLP) systems attempting to process the content: many EHRs in most of the healthcare institutions do not interconnect and therefore makes it cumbersome in relation to acquisition and distribution of patient files and information among the health care professionals. Such fragmentation is not only a threat to the availability of data but it also affects the functioning of NLP systems, which requires working with integrative datasets to improve clinical decision-making Weiskopf Weng 2013 any updates.

Despite the fact that EHR systems consists of a lot of data, the burden of overly structuralization will make it extremely difficult for the relevant data to be captured by an interfered system. Because clinical notes take the form, they require the reading and understanding of the word, of the free text, which makes NLP systems not only constrained in the use of vocabulary and techniques but also the diverse ways language may be applied including childish ways, particular abbreviations and lack of data. Because of such inconsistency in the quality of data, errors in information retrieval will happen and undermine the use of EHR data for clinical decision-making Zhou et al. 2020 updates.

### 2.2 DISRUPTION OF PATIENT-PHYSICIAN INTERACTION

Another area of concern with the growing trend towards the usage of EHR systems remains the diminishment of the interpersonal aspects of with particular attention to the interaction with patients. As EHR usage expands, physicians have been observed taking more time to document patient involvement which in turn, decreases the amount of interaction, which is important in establishing trust and perceptions in healthcare. Research shows that close to 50% of a physician's time is consumed in EHR tasks rather than attending to patients (Downing et al., 2018). As a result, there is less attention paid to the patients and more on data entry as that is what the current model is intended for.

The excessive and careless use of Electronic Health Records is a reason for poor patient outcomes which in turn leads to discontented doctors. Most often, the linking and consideration of the EHR documentation with the patient situation, takes away from the time when the clinician is seeing the patient, which affects the entire exploratory process, which also accounts for a psychoemotional state, non-verbal behavior, and so on (Zhang & Zhang, 2023). All these changes lead to a deterioration of the care quality and the weakening of patients' and physicians' relationships.

## 2.3 Automation Complacency, NLP Inaccuracies, and Algorithmic Bias in NLP Systems

The phenomenon of automation complacency, inaccuracies in NLP and algorithmic bias are also significant obstacles, especially in the conduct of medical NLP systems. Due to the growing use of NLP technology in the healthcare industry, there are concerns of automation complacency among clinicians i.e. depending on automated systems to do most of the work without any output validation. If these NLP model systems cannot grasp the clinical framework within which the data is being used or the intended framework of the patient, on most occasions the results will be falsified or offbeat (Challen et al., 2019). This excessive reliance on technology for generating and prescribing solutions can lead to misdiagnosis in cases that are more complicated.

NLP is inaccurate not just because of algorithms but because of training data that is always incomplete and unstructured. Pressman et al. (2024) explains that clinical data such as clinical notes include inconsistent words, abbreviations and tone of the doctors which leads to errors in processing the data within the NLP systems. Further, algorithmic bias is an important challenge. The data that NLP models get trained on incorporates the biases found in health care data including but not restricted to race, gender and socioeconomic status. The studies of Challen et al. (2019) show that bias inheritance is addictive, and the models trained on that data will only continue to deepen the health disequilibrium among the different groups over time. These biases may contribute to wrong diagnosis and treatment, which can be worse for some patient groups, and hence affect the equitable ability in making clinical decisions.

## 3 Solution

The American Medical Association (AMA) Code of Medical Ethics provides a framework that governs the professional responsibilities of physicians. It emphasizes the importance of key ethical principles such as providing competent medical care, with compassion and respect for human dignity and rights, upholding the standards of professionalism, being honest in all professional interactions, respecting the law and the rights of patients, colleagues, and other health professionals, and safeguarding patient confidences and privacy within the constraints of the law. and caring for a patient, regard responsibility to the patient as paramount. This ethical code ensures that patient care is conducted transparently, fairly, and with respect for individual autonomy. To align with these principles, healthcare systems must employ comprehensive and accurate medical data practices that improve both patient outcomes and ethical decision-making. This necessitates an EHR system and NLP model that both adhere to and enhance these ethical standards.

### 3.1 Designing of New Dataset

In designing a new Electronic Health Record (EHR) dataset which complies with the American Medical Association (AMA) Code of Medical Ethics, a spotlight has been put on minimizing the risks involved in both information inaccuracies and information unethical practices. In this design, each element of the dataset is developed in a way that maximizes the principles of transparency, fairness, confidentiality, and beneficence as provided for in the AMA framework. This new dataset embraces the idea of collecting sufficient and precise information but there is also an emphasis that the information is only used to enhance the health of patients and justice in their practice.

The demographic data of patients is important in this new dataset. The dataset also includes various kinds of demographic data provision so that there is no discrimination in the provision of healthcare including data on race, ethnicity, sex, age, and even social class among others. Demographic information allows for the elimination of health equity bias within population groups and the systematic exclusion of certain individuals from health and medical services. In accordance with the justice principle of the AMA, this component allows all patients to be treated fairly and gives a basis for individual-based and fair medical treatment.

As for the medical history trait, the dataset accounts for each patient's previous maladies, treatments, medicines, files on illnesses or health history in the family. This component aims to ascertain that doctors have complete and accurate patient medical records, which are necessary for making clinical decisions. Given the format of these records which are constantly refreshed, the dataset encourages

openness as well as addresses the legal aspect of obtaining informed consent. The clinical recommendations that physicians make are based on supporting evidence, and patients are made aware of available treatment options and the rationale for such treatments.

One of the most important features of the new dataset is clinical encounter data. Such data documents each time the patients see or communicate with healthcare professionals and includes the type of care given. Each record incorporates the purpose of the visit, tests performed, management of the patient, and clinician's comments. Through this data, it is possible to ensure that each and every medical event which is carried out is properly captured. Access to well justifiable clinical decision and supportive physician's progress notes helps both physicians and patients to provide or seek appropriate care over time. This assists maintaining continuity of care while making sure that patients know why they are treated and bringing in more trust and partnership between the patients and health care providers.

The database also contains an appendix of lab and diagnostics data with the objective of improvement of clinical outcomes through documenting the results of tests and diagnostic methods. This section is paramount to the practice of evidence-based medicine by including supporting data that is current and always verified. In order to improve tracking of interventions and their outcomes, the dataset captures all means of intervention, be it medication, surgery or therapy that are utilized in the treatment process. This section captures the outputs of each modality used in the intervention, both short and long term, assisting in the modification of treatment by the healthcare providers based on the patients' and the clinical outcomes.

The protection of confidentiality is extremely important with respect to this dataset. Access to sensitive personal data concerning patients is restricted to authorized medical staff only through the application of stringent access restrictions. Further, there is a record feature that logs every event of data within the health records when accessed or tampered with so as to adhere and ensure that ethical values regarding patient's confidentiality are upheld.

An important innovation of this dataset is the inclusion of bias mitigation mechanisms. Since various historical datasets are inherently biased, some form of bias surveillance is done on the patient care process, detecting and addressing unequal care across all patient groups. This aspect helps identify and rectify under treatment or over treatment of patients from different races, gender and economic levels. The dataset serves as a means to address treatment patterns where abnormalities are detected and heightened efforts to distribute proper treatment equally throughout the healthcare system.

Apart from things having being updated in real time, there is the learning aspect in the dataset whereby the most recent medical and clinical information is incorporated. This means that the healthcare providers are in a position to make accurate and informed practices since they have the latest knowledge regarding recent scientifically developed practices. The fact that the dataset is updated with experts supporting evidence-based information, the doctor has the right to assist in the enhancement of the quality of the physician actions offered to patients.

## 3.2 DESIGNING OF NEW NLP MODEL

Upstash Vector is an advanced vector database that can scale to address real-time indexing and retrieval of big data. Upstash Vector is different from traditional keyword or relational databases that require searches based on exact keywords since it's built using vector embeddings of data that exist in a conceptual space. Upstash Vector offers a vector database that enables scalable similarity searches across millions of vectors, complete with features such as namespaces, metadata filtering, and built-in embedding models (Dumandag, M, 2024). Thus the system can perform a search even if some of the words are replaced with synonyms referring to the same concept and the meaning is not affected.

Upstash Vector works by embedding data into high-dimensional vectors. Due to the nature of the data, there are relationships between different data points. These vectors are then housed in a database and compared based upon certain distance calculations such as cosine similarity to ascertain how the records relate with each other. When prompting the system, various measurements termed as vectors are retrieved by the query vector and this happens regardless even When prompting the system, various measurements termed as vectors are retrieved by the query vector and this happens regardless even if the exact keywords are not used, the meaningfully related data is returned.

Within our NLP model, Upstash Vector plays a crucial role in ensuring real-time, context-aware data retrieval. Medical data is often characterized by diverse terms used for similar conditions, for example heart attack and myocardial infarction. Ordinary systems that rely on keywords might not capture some records and even if they do, they are likely to default to skimming the surface of the information. That is why Upstash Vector is effective in retrieving all relevant information since it evaluates the relationship that exists between different terminologies. This guarantees that the doctors put across all the relevant information regarding the patients, therefore, minimizing the chances of erroneous diagnosis owing to scanty information. Such real time indexing also means that data such as that of the patient can be written and accessed quickly which is important where treatment needs to be given quickly.

BGE-M3 is a stunning transformer variant which has been particularly significant for building semantic sentence embeddings. BGE-M3 embedding captures the context and meaning of sentences and/or phrases and relies on their contextual rather than their literal word usage to categorize and rank sentences. Quite notably, BGE-M3 is applied on transformer structures that are dominant in the area of NLP such as sentence classification, sentence retrieval, and sentence clustering.

BGE-M3 embeddings are high dimensional vectors that capture the information contained in the text. It incorporates a transformer, which means that even when output is generated word by word, words are never treated in isolation but in their context to other words. Because the model is exposed to a huge number of data points, it is able to perform well on a very wide range of languages. Thereafter these embeddings can be effectively utilized for semantically comparing and retrieving sentences which are not identical but are using different lexical items.

BGE-M3 in our NLP model deals with the understanding and interpretation of clinical data. Different doctors will prefer different terms to describe the same medical concept. For example, "renal insufficiency" and "kidney failure" can be used interchangeably, but keyword-based systems may simply ignore these. BGE-M3 consolidates these terms in the same vector space so that they are all treated as related to the same frame of reference, so any index with such an index will return all documents with or without terms. This is key to enabling clinicians not to miss important patient data, leading to more accurate and better diagnoses for patients.

The coalescence of Upstash Vector and BGE-M3 enhances our NLP model to support fast, semantic, and contextual data retrieval. Upstash Vector possesses the capability for patient record retrieval in clinical systems and other similar large repositories within units of semantically related web pages onscreen. BGE-M3 extends the structure of semantic and dynamic chunks to include the meaning of data by getting records not by keywords but by the sense of them.

With this integration, the quality, and correctness of medical decisions are enhanced to a greater extent. To illustrate, if a physician asks for a coronary heart disease in a patient, due to BGE-M3's semantic embeddings the system will find terms such as "myocardial infarction" or "coronary artery disease" as well. Upstash Vector, however, makes sure that in any case such information is fetched as fast as possible, so there are no delays in critical clinical cases.

Furthermore, this model is particularly beneficial in mitigating the problem of algorithmic fairness within the health care domain. Employing fairness audits and fetching the data according to its context not its history enables all the patients to be treated without discrimination. This is important in safeguarding the core values of the medical profession and endorsing the provisions captured in the AMA Code of Ethics.

### 3.3 EVALUATION AND COMPARISON WITH CURRENT MEDICAL AI MODEL

The medical NLP system that we developed including Upstash Vector for retrieving data that is frequently updated and BGE-M3 for understanding the sense of data is an advance from the current systems of Dragon Medical One and Symptomate. On the whole, conversion of voice to text is done effectively in the case of the Dragon Medical One, but its efficiencies are poor when it comes to the more intricate tasks of data persuasions and analysis of latent meanings in the given material. It depends primarily on keyword searches and this criticism can be directed towards it due to its overreliance on a few keywords such that some patient's important information may be lost when other similar words or phrases are used to refer to that particular condition. Take for instance, the situation in which a doctor records the history of a patient using the word "myocardial infarction"

while another writes "heart attack." Most likely, dragon medical one is not going to retrieve both records. In short, either of them will not be shown to the clinician, so between the records, there is a tendency to have incomplete data presented to the clinician. Moving swiftly, within this model, the patient can be searched accurately and contextually. This BGE-M3 applies even more finely as after converting things to high dimensional vectors using deep learning, it provides meaning to the terms searching that 'myocardial infarction' and 'heart attack' are interchangeably hence in a search English they are bound to produce similar results. So thorough retrieval of all the latter encyclicals which the Dragon Medical one cannot do.

There is additionally a gap in Dragon Medical One on how to assess and deal with bias. It does not cognize or control for demographic stereotypes in the underlying medical data. This could reproduce inequities in healthcare. In our case, however, such bias was and still is regularly monitored to recommend adequate and unbiased resources among the target groups. This is important in preventing biased medical decisions on data where the patient care is historically based on prejudice to a particular race or demographics.

Symptomate is an example of another system often referred to as that which here allows a rapid diagnosis via symptoms, but does not allow for more complicated patient clinical histories. Symptomate works within the constraints of mapping given symptoms with predefined diseases and does not consider many other factors in the patient's history such as other diseases. For instance, a patient with complaints of kidney problems may have diabetes under control. In this case, Symptomate would not be able to determine its influence on the underlying diseases in regards to further imaged-based diagnostic proposals. In that regard, our NLP model is different because data from all sources including lab results, treatment history, and past diagnosis which makes it offer recommendations that are more prevalent in the context. Through BGE-M3, for instance, the model is able to relate and integrate conditions that are related for prevention measures when coming up with the recommendations considering their interconnectedness providing the clinicians with a complete picture of the patient health status.

Symptomate also performs poorly in this regard as it does not allow for data retrieval and provision of support in real-time. Instead, Symptomate provides an analysis of selected symptoms in relation to a fixed set of symptoms which it has awful performance in erecting and accessing the EHR for the purpose of rendering contemporary clinical decision support. The adoption of Upstash Vector by our model makes it possible to access patient data which makes certain the decisions made by clinicians are the most recent and pertinent. This functionality is particularly critical in specific situations involving biphasic decision-making such as emergency cases, where patient outcome depends on the provision of the right information.

## 4 CONCLUSION

In conclusion, this paper demonstrates how the combination of Upstash Vector and BGE-M3 in a novel dataset and a new NLP model in the framework of ama ethics code will help to break the constraints of the present medical data systems by providing timely and relevance-oriented searching, comprehension of the semantics and elimination of biased prejudice. Such improvements guarantee that healthcare professionals can get quite complete and precise data of patients, helping them to make sound decisions.

Further work should be oriented to the formulation of complete AI guidelines in the context of health care, focusing on training for both the model and users, and performing constant bias evaluation, establishing collaboration between human and AI, and making the model practical in order to make it appropriate for use worldwide.

## 5 LIMITATIONS

Incorporating Upstash Vector and BGE-M3 into the function of our NLP model presents positive pay-offs particularly in improving healthcare data management in terms of accuracy fair and personalized management of records. However, quite a number of issues remain particularly regarding the sensitive nature of medical data', the need to avoid over-dependence on AI systems by health practitioners, and the AMA principle's limitations. The AMA ethics code, though useful, are too

broad and abstract to assist with moral difficulties that arise in the process of integrating AI in medicine. Therefore, we need a more detailed and structured ethical framework that can address both technical and ethical challenges posed by AI-driven healthcare systems.

## 5.1 THE HIGH STAKES OF MEDICAL DATA

While non-medical datasets such as Wikipedia may have inaccuracies that are not serious in nature, medical data is of a very high level of sensitivity and such inaccuracies can't be tolerated. The Information Age allows a lot of convenience such as rapid retrieval of medical data, accurate interpretation of the data and decision making is also made easier aided by AI, but this technological advancement does have its drawbacks. In medical AI, anthropocentric accuracy not simply applies but the direction and internal contents should be regulated in more detail.

Consequently, we believe that any NLP model has to be equipped with strong auditing mechanisms to maintain data quality and avoid mistakes. Again, there remains a need for regular auditing of data so that the issue of whether the system is able to retrieve and comprehend the data gathered is adequately addressed. Considering the differences in medical language and unstructured data in clinical notes, these audits have to look for such anomalies and remediate them as they occur, or clinical discrepancies will likely occur. This is particularly important because making clinical decisions using incorrect or incomplete information can affect a patient's wellbeing and health.

## 5.2 TRANSPARENCY AND ACCOUNTABILITY IN AI

Ensuring transparency and accountability in the processes of AI Based decision making systems is another issue that hinders ai implementation in healthcare systems. Healthcare practitioners such as physicians, have to have faith in the recommendations made by various NLP models, however they also have to appreciate the reasoning behind such recommendations. The NLP model has to be able to ascertain what text came from where, and provide proportional reasoning as to why the AI made a particular recommendation etc.

Also, there is a need for coherence in data provenance, ensuring that the healthcare providers are aware of the sources of the data, how it has been processed, and the purpose for which it was collected. It's important to know that medical documents include much more than clinical notes or lab results, and may encompass imaging studies, there is a lot that requires transparency in dealing with such varied data. It is also key in building in transparency and accountability mechanisms to enable the AI systems to be operated in a manner that is trustworthy and adheres to clinical governance thereby protecting patients.

## 5.3 BIAS MITIGATION AND AMA FRAMEWORK

While playing by the rules set in place in the AMA Code of Medical Ethics within a business environment has been embraced by many organizations, it is just too basic and theoretical in its application, and therefore seeks for the integration of AI in healthcare for better approaches. In particular, the considerations of the AMA are rather addressed at a more abstract level and contain broader principles - respectful, helpful, professional – which are very important, however, not very detailed towards the development of medical AI. As with the healthcare system, there is an increasing role for ai technologies, and hence more effective and practical frameworks to constrain the use of AI to healthcare have to be developed.

Algorithms are always biased towards a particular trend amongst those societies despite the fact that ai is intelligent. These claims are even more compelling in admitting that there are some people manning the category of 'Physician bias'. This concern partly stems from the generic nature of the AMA objectives. Nonetheless, our Interpret analysis or how recommendations are made to patients entails fairness audits and applies to our NLP model. This is within the context of the audit objective. Each of these audits seeks to protect particular features by ensuring that there is no excess or insufficient care or accurate diagnosis to a particular group.

Yet, preserving the benefit of bias mitigation is much harder in the working AI systems. Systems should be designed with monitoring and enable updates. The reason for this is biological; bias does not reside in healthcare data permanently; therefore, the system should be flexible to changes

so that justice is experienced by all patients despite medical decisions taken by the system. Bias Mitigation features will be useful for our model in that they will ensure that systemic healthcare differences especially on distribution are not encouraged and the ethical dimension will be improved even surpassing the provisions of the AMA Code.

### 5.4 OVER-RELIANCE ON NLP AND THE NEED FOR HUMAN OVERSIGHT

The healthcare sector, where artificial intelligence is being incorporated at a high rate, is facing one of the unintended challenges, overdependence on such systems by doctors. The NLP models like our Search, significantly improve access to data and elements of decision making, however, there are concerns that clinicians may become overly reliant on the system's suggestions, a phenomenon known as automation complacency. Patients may suffer from the negative consequences of this over reliance where when doctors are to emphasize on their recommendations, for example in an AI buttressed cased, patients may be considered 'wrong' in the absence of adequate explanation, thus naturally leading to a wrong diagnosis and wrong treatment.

In addition, this undue dependence can also lead to deterioration in the level of quality doctor-patient's relationship which is still an integral component of medical practice. Healthcare, a domain confined to "human" language, acknowledgement, and comfort has no salvation in any level of AI. None with the most ingenious design and advanced technologies. Administrative functions and analytics of AI can reduce some chore but not the least activities of physicians with patients that include relation and reasoning.

There is a risk of over-dependence on the use of AI. As such, there should be on-the-job training that includes training on the right application of NLP systems with regard to human management. Education should enable health care professionals to evaluate the relevance of artificial intelligence suggestions that AI is an instrument rather than a replacement of judgment within the clinical outlines. This may help physicians avert increasing their automation complacency and encourage them to participate in the decision-making activities.

Moreover, training should aim at enhancing how patients and doctors communicate by using AI for those interactions where more human attention is less helpful. AI-enabled technologies can make routine processes carry less importance which gives room for the doctor to spend more time listening to patients, outlining their treatment, and addressing their worries. In this way, the reason why most people fear artificial intelligence will be addressed: AI will enhance care as opposed to diminishing it, provided its usage is done right and with enough human management.

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
