# OpenReview forum: "Enhancing Medical NLP Systems: Integrating Upstash Vector and BGE-M3 for Accurate and Ethical Healthcare Data Management with Reduced Bias"
_ICLR.cc/2025/Conference — Submitted to ICLR 2025_

### Official Review · Reviewer_34wG · 2024-10-30

**Soundness:** 1
**Presentation:** 2
**Contribution:** 2
**Rating:** 3
**Confidence:** 4

**Summary:**

This paper proposes a novel medical NLP system that combines Upstash Vector for real-time data retrieval and BGE-M3 for semantic understanding, while incorporating the AMA Code of Medical Ethics principles. The system aims to address several critical challenges in current medical NLP systems, including fragmented data, disrupted patient-physician interactions, and algorithmic bias. The authors present a new dataset design and NLP model that emphasizes ethical considerations, bias mitigation, and improved semantic understanding of medical terminology.

**Strengths:**

Comprehensive Problem Analysis
    • Clear identification of current challenges in medical NLP systems
    • Well-structured presentation of limitations in existing systems like Dragon Medical One and Symptomate
    • Strong emphasis on ethical considerations and bias mitigation
Novel Technical Integration
    • Innovative combination of Upstash Vector for fast retrieval and BGE-M3 for semantic understanding
Ethical Framework
    • Strong incorporation of AMA Code of Medical Ethics principles
    • Detailed consideration of patient privacy and data security
    • Emphasis on maintaining doctor-patient relationships
Practical Considerations
    • Recognition of real-world implementation challenges
    • Thorough discussion of limitations

**Weaknesses:**

Empirical Validation
    • No quantitative evaluation of the proposed system
    • Lack of comparative performance metrics against existing systems
    • No experimental results to support claims of improved accuracy
Implementation Details
    • Insufficient technical details about the integration of Upstash Vector and BGE-M3
    • Limited discussion of computational requirements and scalability
    • Unclear specification of the training process for the NLP model
Cost and Resource Considerations
    • No discussion of implementation costs
    • Limited analysis of required computational resources
    • No consideration of training requirements for healthcare providers
Generalizability
    • Limited discussion of cross-linguistic applications
    • Unclear applicability to different healthcare systems
    • No consideration of international medical standards beyond AMA

**Questions:**

• How does the system perform in terms of processing speed and accuracy compared to existing solutions? What metrics are used to evaluate this?
• What are the specific training requirements for healthcare providers to effectively use this system while avoiding over-reliance?
• How does the system handle multi-language medical records and cross-cultural medical terminology?
• What is the estimated cost of implementation for a typical healthcare facility, including both technical infrastructure and training?
• How does the system handle edge cases where medical terminology is highly specialized or novel?

---

### Official Review · Reviewer_GcsC · 2024-11-04

**Soundness:** 1
**Presentation:** 1
**Contribution:** 1
**Rating:** 3
**Confidence:** 4

**Summary:**

The paper presents a NLP system for electronic health records (EHRs) which combines text embeddings from BGE M3 with vector database Upstash vector. The resulting system is promised to allow efficient and semantic data retrieval. The authors generally discuss shortcomings of existing systems and outline the merits of the proposed system.

**Strengths:**

The research direction is important, as it advances AI usage for health care, allowing to better treat patients by giving medical professionals more readily available information.

**Weaknesses:**

- The paper lacks a clear, formal description of the proposed system. It is not discussed how the data is processed, how the embeddings might interact, what is eventually shown to the medical professional
- The paper lacks an en empirical or theoretical evaluation. No datasets have been chosen to compare the proposed systems with any baseline.

**Questions:**

- Have you implemented the methods and tested it on available benchmark datasets to assess its performance? If so, what are the empirical results in comparison to a baseline system?
- In that sense, what is the empirical added value of combining the two components, i.e., embedding database and transformer-retrieved embeddings?
- Please curate a list of main competing scientific approaches and discuss how they are conceptually different in a designated section.

---

### Official Review · Reviewer_PBmH · 2024-11-09

**Soundness:** 3
**Presentation:** 3
**Contribution:** 3
**Rating:** 6
**Confidence:** 4

**Summary:**

This paper works to present a proposal based on a specialized NLP model focused on a serious need for change in how healthcare data is managed, particularly in the area of Electronic Health Records (EHR) systems. It targets a few very highly common issues of the current medical NLP systems, such as fragmented data, algorithmic bias, limited interoperability, and ethical complexities in data handling. To do so, the authors make a model proposal within which two important technologies are embedded: Upstash Vectors and BGE-M3. It uses Upstash Vector as a high-dimensional vector database in order to enhance real-time, contextually relevant data retrieval using similarity searches over patient data. This component overcomes the limitations of keyword-based searches by allowing the retrieval of related medical terms and conditions, even when synonymous or contextually similar phrases like "heart attack" and "myocardial infarction" are used. This will allow physicians to have all the important patient data at hand in a timely manner, which is crucial in making an accurate diagnosis and for timely treatment, especially in urgent care scenarios. The BGE-M3 transformer model is integrated into the system in order to enhance the system's understanding of clinical language at a deeper semantic level as it parses variations in terminology and captures the context in which terms are used and as such, enables the system to identify and associate different ways that clinicians may describe similar medical conditions, increasing the consistency and comprehensiveness of information retrieved.

The paper puts this model in the context of the American Medical Association (AMA) Code of Medical Ethics in order to ethically handle aspects of health care data management, guarantee transparency, confidentiality, and accuracy in the handling of patient information. As such, the model indeed includes bias mitigation strategies and fairness audits to avoid algorithmic bias, specifically biases due to race, gender, and socioeconomic status, biases that are commonly mirrored in healthcare data and are observed as such in other research on the topic. These mechanisms will dynamically monitor and correct any potential discrepancies in the data used in decision-making to ensure equitable treatment for diverse patient populations. It then lists how current NLP models fall short compared to their approach, stating the example of Dragon Medical One which is good at speech-to-text tasks but relies too much on keyword searches that considerable retrieval of data is incomplete or inaccurate if terms are varied. Symptomate is also limited by the symptom-based diagnosis method and is not amalgamated with a patient's entire clinical history. In sharp contrast to this, the proposed model not only gains access to a more panoramic aspect of patient data through both historical and real-time information but also refines data interpretation accuracy using semantic embeddings.

The paper also goes on to discuss some of the limitations intrinsic in relying entirely on the AMA Code of Medical Ethics by pointing out that although the code provides a moral basis, it is too nonspecific to guide all types of AI-based medical decisions effectively. The autors of the paper also call for the development of more practical and detailed ethical frameworks to address challenges arising uniquely from the integration of AI in healthcare. In the final analysis, this model, by bringing together the state-of-the-art retrieval capabilities of Upstash Vector and the semantic depth of BGE-M3, points toward a significant improvement over current existing systems. This integrative approach is purported to ensure accurate, bias-reduced, and ethically guided data handling with applications that could improve healthcare delivery by letting clinicians make informed data-backed decisions in real time.

**Strengths:**

I would describe the paper’s strengths as its original, high quality, clear, and significant contribution to the field:
The novelty of the paper is in the fact that, for the first time to the best of my knowledge, these two technologies, Upstash Vector and BGE-M3, have been applied in a healthcare-oriented NLP model in this way. This is quite an innovative combination since it integrates real-time data retrieval with semantic interpretation, both important components in managing the complexity of medical records. In addition, active consideration of ethical concerns in the handling of data and decision-making through explicit inclusions of the AMA Code of Medical Ethics and fairness audits represent a unique approach that distinguishes this model from many existing NLP systems currently applied in healthcare. The quality of the paper is also quite high, especially for the design of the model and the methodological approach to problem solving. Clearly, by choosing such components as Upstash Vector and BGE-M3, the technical capabilities of the model are in line with what is expected from the healthcare field, such as addressing topical issues like interoperability and bias in clinical data. However, the research needs more empirical validation, more details related to benchmarking, and comparative metrics against leading NLP models in the healthcare space. The clarity of the paper is acceptable too as explanations for each component of the proposed model have been presented in a well-structured manner, technical terms and processes are usually well-defined, and placing the research within the broader context of medical NLP and ethical AI is clearly delineated. However, more complex aspects like bias mitigation and the actual implementation of fairness audits could, with additional explanations, be even easier to comprehend for the reader. The significance of the paper is also not something to overlook as it addresses some of the most important challenges of medical NLP systems. The work should be of particularly high value to healthcare professionals since AI-driven decision-making has significant implications for patient outcomes and equity in the applications of AI in this domain. This study works to contribute to the next wave of medical AI by putting forward a real-time, NLP model with considerations toward its accuracy and ethics-a contribution that could bring more trust and fairness into AI-driven healthcare solutions. It offers a potentially transformative solution that might further inspire research on AI ethics and bias mitigation for other sensitive sectors.

**Weaknesses:**

A lack of empirical validation may be one of the most significant weaknesses in this paper as it does not have any empirical evidence to back up the claims about the model. While there is a theoretically good support for why the model is functional, what it lacks is test results or metrics that are necessary to establish how far integration of Upstash Vector with BGE-M3 will progress toward improved data retrieval, reduced bias, or increased semantic understanding in a clinical setting. Quantitative comparisons with established NLP models, such as Dragon Medical One and Symptomate, on standardized health datasets give significantly more value to these claims. In this view, the authors should therefore benchmark quantification on improvement in retrieval efficiency of data, retrieval accuracy across varied terminologies, and reduction in detected bias. Also, a lack of information on mechanisms for reducing bias is also present with the paper insisting that fairness audits and bias mitigation are main elements, but it does not reveal much about how such mechanisms are implemented in the model -- Specific details with respect to the process of how this fairness audit is done would be helpful: for example, what kind of biases it ensures the model does not have (racial, gender, socioeconomic), how frequent such audits are, and what the steps are to identify and rectify the bias are all things that elucidate how reduction of bias is actually achieved in this model.  As such, this work would be stronger if more comprehensive methodology were provided for detecting and correcting bias within the NLP model. A shortage of concrete examples and case studies and specific examples or case studies also persists, with more such examples being needed to help elucidate how the model actually works. These could include, for instance, a case study of how Upstash Vector retrieves relevant but variably labeled medical data-"myocardial infarction" and "heart attack," for example so that through such examples, the model can exemplify ways to better present information to clinicians. Examples of this are showing where BGE-M3 consolidates synonymous terms into one term in order to lessen the instances of errors in retrieval; this provides the readers with some insight into the model's practical advantages and would make the proposed improvements more tangible and relevant to real applications in healthcare. Last but not least, it would also be great if issues of scalability and computational resource requirements would be addressed and considered – the autors should consider including optimizations that may make the results more broadly accessible as these will highlight the contributions of the paper in a much clearer and actionable manner, thus making it more compelling and strengthening its contribution as a source in the area of AI-driven healthcare.

**Questions:**

Could you please elaborate on the specific methodologies employed for fair audits and bias mitigation? As in, are there particular techniques or metrics you resort to while detecting and measuring biases in patient demographics, race, gender, and even socioeconomic status?

Is there some periodic reassessment of the model concerning new biases, and if so, with what frequency?

As your solution is claimed to outperform current NLP systems, could you please provide some quantitative results regarding your model compared to current NLP systems in terms of data retrieval accuracy, speed, and bias reduction? Even preliminary comparisons to a benchmark health care data set would substantiate claims of improved retrieval and reduced bias brought forth by the model. If experiments are not possible for the submission, can you describe empirical tests that are planned, including evaluation metrics or specific datasets you would use?

If possible, please provide specific instances to demonstrate the model's usefulness in undertaking critical clinical cases in real time due to the incorporation of Upstash Vector and BGE-M3. How would it retrieve data for a patient with overlapping symptoms like "chest pain" and "myocardial infarction" from different types of EHR entries? Examples of how the model detects and corrects for prospective bias in patient treatment recommendations would also help in better defining the real-world applications and benefits of your novel model proposition.

In the context of automation complacency concerns in healthcare, can you please provide more details on how your model either detects or lessens that concern? Are there identified strategies, such as confidence levels or alerts, which would notify a clinician that AI-generated recommendations may need further verification? Would guidelines, such as requirements that clinicians double-check AI insights, or any training recommendations to avoid over-reliance on AI outputs, be something you would consider?

With the integration of Upstash Vector and BGE-M3 for real-time retrieval and semantic understanding, the computational overhead rapidly grows huge. Can you please provide a general idea of the computational requirements of your model? Can such a model be deployed in a healthcare facility with relatively smaller resources? Where scalability might have been a factor in the design of your model, please describe any optimizations done to lower resource utilization, including but not limited to memory footprint, processing speed, and/or storage.

While the AMA Code provides a core ethical structure, I would like to understand whether additional, more AI-specific ethical guidelines will be considered for integration. For example, are IEEE, WHO, or other guiding bodies on AI ethics helpful in providing more holistic recommendations?Are there any important ethical areas of concern in your model, like patient privacy in real time access to data, for which you have taken extra measures beyond the guidance of the AMA?

---

### Official Review · Reviewer_uBzA · 2024-11-10

**Soundness:** 1
**Presentation:** 1
**Contribution:** 1
**Rating:** 1
**Confidence:** 5

**Summary:**

The authors outline several issues in the current era of electronic health records and how these affect AI technologies in healthcare. They then propose a new design of both electronic health record data and a model that leverages vector databases and contextual embeddings fast, semantic, and contextual data retrieval. They describe how two existing systems (Dragon Medical, Symptomate) fall short/

**Strengths:**

The paper nicely overviews overarching weaknesses in the current generation of AI + NLP technologies and electronic health record (including bias, data fragmentation, missingness, automation bias).

The authors outline in detail possible limitations of their proposal.

**Weaknesses:**

While the paper proposes a new path forward in terms of data and model, there is no information about the implementation or quantitative results about the setup. As a result, it reads more as a position paper.

The authors have not contextualized their proposals in related work, e.g. other literature that deals with semantic retrieval in electronic health records, e..g “I Don't Have Time to Dig Back Through This: The Role of Semantic Search in Supporting Physician Information Seeking in an Electronic Health Record."

**Questions:**

Has the described system actually been built? It was described in the present tense, but no implementation details were included.
Are there any quantitative metrics that can back up the strong claims made in the paper, such as "it guarantees no unfair treatment of patients belonging to different groups"?

---

### Meta-Review · Area_Chair_SSEt · 2024-12-21

**Metareview:**

The paper argues for adoption of a transformer variant called BGE-M3 and an embedding method Upstash Vector to improve healthcare data management. They describe how two existing systems (Dragon Medical, Symptomate) fall short of the desired characteristics for medical systems.

Strengths:
 - Describes the weaknesses of current medical data management and clinical systems well

Weaknesses
 - No empirical validation, no proposed methodology, and no supporting evidence for claims
 - Lack of contextualization for arguments within related work
 - Contributions to the ICLR community are unclear, making this perhaps not the right venue (as opposed to a medical journal)

Based on the overwhelming lack of justification for the arguments of the paper, I am recommending reject.

**Additional Comments On Reviewer Discussion:**

Reviewers were unanimous in their decision and identification of weaknesses in the paper.  Note that Reviewer PBmH's review is likely to be LLM generated due to the verbosity in summarizing the paper and lack of direct analysis and was disregarded.

---

### Decision · Program_Chairs · 2025-01-22

Reject